# Changes in EEG Recordings in COVID-19 Patients as a Basis for More Accurate QEEG Diagnostics and EEG Neurofeedback Therapy: A Systematic Review

**DOI:** 10.3390/jcm10061300

**Published:** 2021-03-22

**Authors:** Marta Kopańska, Agnieszka Banaś-Ząbczyk, Anna Łagowska, Barbara Kuduk, Jacek Szczygielski

**Affiliations:** 1Department of Pathophysiology, Institute of Medical Sciences, Medical College of Rzeszow University, 35-959 Rzeszow, Poland; 2Department of Biology, Institute of Medical Sciences, Medical College of Rzeszow University, 35-959 Rzeszow, Poland; agnieszkabanas@o2.pl; 3Student Research Club “Reh-Tech”, Medical College of Rzeszow University, 35-959 Rzeszow, Poland; alagowska7@gmail.com (A.Ł.); barbarakuduk@wp.pl (B.K.); 4Department of Neurosurgery, Institute of Medical Sciences, Medical College of Rzeszow University, 35-959 Rzeszow, Poland; 5Department of Neurosurgery, Faculty of Medicine, Saarland University, 06841 Saarbrücken, Germany; jacek.szczygielski@vp.pl

**Keywords:** QEEG, neurofeedback, COVID-19, EEG record, EEG neurofeedback training, neurological diseases

## Abstract

Introduction and purpose: The SARS-CoV-2 virus is able to cause abnormalities in the functioning of the nervous system and induce neurological symptoms with the features of encephalopathy, disturbances of consciousness and concentration and a reduced ability to sense taste and smell as well as headaches. One of the methods of detecting these types of changes in COVID-19 patients is an electroencephalogram (EEG) test, which allows information to be obtained about the functioning of the brain as well as diagnosing diseases and predicting their consequences. The aim of the study was to review the latest research on changes in EEG in patients with COVID-19 as a basis for further quantitative electroencephalogram (QEEG) diagnostics and EEG neurofeedback training. Description of the state of knowledge: Based on the available scientific literature using the PubMed database from 2020 and early 2021 regarding changes in the EEG records in patients with COVID-19, 17 publications were included in the analysis. In patients who underwent an EEG test, changes in the frontal area were observed. A few patients were not found to be responsive to external stimuli. Additionally, a previously non-emerging, uncommon pattern in the form of continuous, slightly asymmetric, monomorphic, biphasic and slow delta waves occurred. Conclusion: The results of this analysis clearly indicate that the SARS-CoV-2 virus causes changes in the nervous system that can be manifested and detected in the EEG record. The small number of available articles, the small number of research groups and the lack of control groups suggest the need for further research regarding the short and long term neurological effects of the SARS-CoV-2 virus and the need for unquestionable confirmation that observed changes were caused by the virus per se and did not occur before. The presented studies described non-specific patterns appearing in encephalograms in patients with COVID-19. These observations are the basis for more accurate QEEG diagnostics and EEG neurofeedback training.

## 1. Introduction

An electroencephalogram (EEG) and a quantitative electroencephalogram (QEEG) are the methods of assessing the electrical activity of the cerebral cortex [1]. Spatial and temporal alterations in neural voltages recorded in the form of classical EEG waves may be further analyzed as to the percentage of different frequency spectra of electrical activity. This technically more sophisticated version of the EEG is called a quantitative EEG because it requires the quantification of records from single EEG sensors by specialized software. Both the EEG and the QEEG, but mostly the QEEG, are important diagnostic tests providing reliable information about brain dysfunction and are inevitable in the diagnosis and monitoring of epileptic seizures, for example. A QEEG is a safe and non-invasive method where the record from the EEG sensors is quantified by specialized software. The results are compared with the normative databases that contain records of healthy individuals in a given age group. In the field of clinical research, this method is often implemented as a measure of the effectiveness of a given therapy/intervention based on the test-retest method [2,3].

The EEG test and the information obtained from the QEEG can be interpreted and used as a clinical tool to assess the functioning of the brain, to predict the effects and to diagnose diseases including the distinguishing of their subtypes. QEEG processing techniques and the use of modern analytical software for EEG/QEEG processing provides us with the opportunity to monitor dynamic changes that take place in the brain during cognitive tasks. The examination is performed in order to detect the cause of the disorders, their nature and to select the most appropriate individualized therapeutic protocol according to the patient’s condition [4,5].

An EEG/QEEG offers the unique opportunity to observe how different pathogens, including the SARS-CoV-2 virus, affect the human nervous system and interfere with the function of the neuronal network during the disease and also to state whether these changes in the brain are only transient or rather leave a stigma even after recovery from infection [6,7,8,9].

This chance to gain knowledge about the exact effects of the virus on the body as well as about its entire spectrum of action may be taken in order to facilitate the diagnosis and treatment of SARS-CoV-2 related disturbances of brain function. Thus, the EEG and the QEEG may become in the near future crucial for the diagnosis and treatment of subsequent neurological sequelae after the transition of the COVID-19 disease. Moreover, the QEEG is a basic component of therapeutic methods including QEEG biofeedback. In this method, coaching the patients to influence their EEG frequencies allows the mitigation of the symptoms of some behavioral disturbances including dementia [10], autism spectrum disorders [11] and attention deficit hyperactivity disorder [12]. Thus, accurate QEEG diagnostics may create a solid basis for considering the regular use of EEG biofeedback therapy as an individual treatment form for those patients in whom neurologic COVID-19 sequelae closely resemble the symptoms of the above-mentioned conditions where QEEG neurofeedback has already been successfully used.

## 2. Objective

The aim of this study was to review the recently released research on changes in EEG records (as a basis for further QEEG diagnostics and possible EEG neurofeedback training) regarding the patients who underwent a COVID-19 infection.

## 3. Literature Review

Publication selection methods. A systematic review of the literature is considered to be the method of integrating scientific evidence, which uses an explicit protocol for the identification, selection and analysis of data qualified for the review. Its purpose is to minimize bias and strives to obtain credible and reliable scientific evidence. For that purpose, we searched PubMed, Google Scholar, Science Direct and the ISI Web of Knowledge databases. The literature study was based on the following keywords combined with the “and” operator in various configurations: COVID-19, SARS-CoV-2, EEG, QEEG, training neurofeedback, EEG biofeedback, nervous system, encephalopathy, epilepsy, brain, virus. A total of 17 articles were analyzed. The literature review covered the last seven months back to October 2020. The relatively narrow time frame of searching was a consequence of the short time of existence of this virus and little scientific research in this field. The stages of searching in databases and the key words applied in the review are presented in Figure 1.

When selecting literature, the individual items, case reports, case studies and systematic reviews were taken into account as well as research in which the EEG/QEEG study was conducted in patients with COVID-19. Additionally, studies on neurological changes caused by the SARS-CoV-2 virus in people who had no neurological/mental illness prior to COVID-19 were considered. On the other hand, the analysis ruled out research on the SARS-CoV-2 virus with no relation to nervous system dysfunction. In addition, the exclusion criteria covered EEG/QEEG tests where the patient did not have COVID-19 and people under 18 years of age (Figure 2).

## 4. Description of the State of Knowledge

Literature analysis. The following aspects of the assessment were distinguished in order to characterize the eligibility criteria: the description of the study group, SARS-CoV-2 infection, EEG/QEEG study, the duration of therapy and a description of the obtained results (Table 1).

## 5. Results

A general review of the literature on the influence of SARS-CoV-2 on the nervous system clearly demonstrates that this pathogen is able to cause diverse neurological symptoms such as headaches, disturbances in consciousness and concentration and a reduced ability to sense taste and smell. These symptoms often persist for prolonged periods after recovery, suggesting that SARS-CoV-2 may cause permanent damage to certain areas of the central nervous system (CNS) or peripheral nerves [24,25,26,27].

Much of the research on this topic indicates that patients suffering from COVID-19 may develop encephalopathy, manifested as a slowing down of thinking processes and memory impairment or personality changes, impaired concentration and sleep disorders [27,28].

Among the studies carried out so far, the patients with the neurological impairment demonstrating these features of encephalopathy usually underwent EEG testing. In one of the reports, specialists from the Institute of Neurological Sciences of Bologna presented the EEG results of a group of 15 COVID-19 patients in the age range from 47 to 79 years (mean age 64.6 years) with suspected encephalopathy. Notably, in all of these cases the EEG recordings were abnormal. Here, most patients displayed a slowing activity in the 4 to 8 Hz range with focal theta or delta waves observed predominantly over the frontal or central region. In 10 cases, no reactivity to external stimuli and no opening and closing of the eyes was documented. Although this study was performed with a very limited number of subjects, it indicated the need for EEG diagnostics in patients with COVID-19 who manifested neurological symptoms as the important tool, assessing the negative effect of SARS-CoV-2 on the function of the central nervous system. The presence of neurological symptoms also justified an indication for the use of more sophisticated diagnostic methods such as the QEEG. More so, the QEEG in COVID-19 patients even if used initially as an assessment tool, may create a solid background to conduct neurofeedback therapy aiming at a complete recovery to a state before the infection [13]. Another report presented by the International Federation of Clinical Neurophysiology displayed two cases of COVID-19 patients (men aged 37 and 42) in whom the EEG test showed a new, unprecedented wave pattern. Here, continuous, slightly asymmetric, monomorphic, two-phase, slow delta waves with a greater amplitude in both frontal regions was described. These waves showed no reactivity to auditory or nociceptive stimuli. In particular, the latter report prompted us to conduct further electroencephalographic studies to confirm whether SARS-CoV-2 may actually be directly involved in producing this unique pattern on the electroencephalogram [14]. Further on, a research group of the Department of Neurology and Clinical Neurophysiology in London analyzed the EEG results of 19 patients with COVID-19 at the age of 37 to 69 (median age of 52), 13 of whom were diagnosed with severe encephalopathy possibly causing a coma after he discontinuation of sedation. A relatively high frequency of the rare alpha pattern was observed in the electroencephalogram with a suggested background of direct SARS-CoV-2 neurotropism [16]. The importance of EEG monitoring in patients with COVID-19 at a high risk of encephalopathy was underlined by the study in which EEG monitoring was performed on five critically ill adult patients with changes of mental state or with seizures (in the 37–60 age range where the median age was 40). In all of these cases, non-specific markers of encephalopathy, diffuse slowing down and rhythmic delta activity were documented [19]. However, encephalopathy with slowing down the EEG activity is not the only pattern of SARS-CoV-2 related electrophysiologic changes. Even more importantly, COVID-19 is also capable of causing epileptic fits. However, it should be taken into consideration that the number of studies on this subject conducted so far is still very limited [20,21,29]. One example here was a small case series preliminary report, which presented case reports of 26 patients aged 30 to 83 where the median age was 64 with 8-channel EEGs. Twenty patients were positive for COVID-19. The EEG test was performed on subjects who experienced changes in their mental state and motor discharges resembling epilepsy. The results of the study showed dominant sharp frontal waves and changes indicating a new encephalopathy [18]. Further on, a single case report published by the International Federation of Clinical Neurophysiology described the case of an 80-year-old female patient suffering from COVID-19. It was accompanied by several symptoms including a restless mental state with altered consciousness. In addition, she developed focal epileptic seizures that resolved after the administration of appropriate pharmacotherapy; however, disturbances of consciousness remained. This patient underwent several EEG tests, the results of which indicated a slowdown in basic activities and an epileptic status manifested in the frontal area. Further on, a three-phase wave activity was found in the electroencephalogram. Generally, these features accompany respiratory failure. Interestingly, in this particular case the neurological deterioration took place despite a stable respiratory status. Consequently, the authors of the study concluded that the development of the three-phase wave pattern was not related to the pulmonary status. They argued that the EEG changes were indicative of a progressive neurological process that was possibly related to direct neurotrophic properties of SARS-CoV-2, underlining the need for an appreciation of COVID-19 as a cause for cerebral electrophysiologic disturbance manifested by a changed EEG pattern [17].

The studies described above were carried out in single or too few patients to draw any statistical conclusion [29] and therefore many questions about the reason of the observed EEG changes need to be raised, implying also that a more elaborate QEEG workup should be performed; for example, in which the apparent abnormalities in the EEG would be described in a more precise, quantitative manner. To date, only one study was found to be implementing quantitative electroencephalography (QEEG) to determine the characteristics of encephalopathy in patients with COVID-19. This examination was, however, performed not during the acute phase of the disease but after the patients had left the intensive care unit (ICU). At that time, the basic EEG test showed an almost physiological pattern with the presence of only a few changes that could indicate the presence of encephalopathy while the QEEG test was much more precise, allowing the determination of the type of (presumably SARS-CoV-2 related) encephalopathy [15].

Certainly, for more reliable evidence of the impact of COVID-19 on electric brain activity, a study in a larger population of patients including those with and without SARS-CoV-2 infection (as a control group) would be mandatory. An important attempt to do so was included in a review of Antony et Haaref who analyzed data from 617 patients with EEG scores reported in 84 studies (the age range was 45–69 and the median age was 61.3 years) including SARS-CoV-2 infected subjects. The authors found that EEG abnormalities were common, mostly in the frontal lobe/area and included focal background slowing, intermittent discharges and rhythmic delta activity. This suggested that non-specific encephalopathy is one of the most common nervous system abnormalities associated with COVID-19. In addition, epilepsy-like discharges (less commonly, seizures or status epilepticus) were observed. These results implied that frontal EEG patterns were a characteristic symptom of COVID-19 related encephalopathy [23]. However, for the valid statement in this matter, it cannot be ruled out that the EEG test is not accurate enough so it may again be suggested that a QEEG should be performed on a larger number of patients to confirm the relationship of COVID-19 with the occurrence of neurological disorders.

If such studies confirmed the coexistence of COVID-19 related changes with symptoms that are possibly amenable to QEEG training, it would also justify the possibility of introducing QEEG-based neurofeedback therapy in these patients. There is a great need for further research on the therapeutic potential of this method in the COVID-19 affected population. What should be investigated is what the QEEG test truly shows in a vast group of COVID-19 patients, whether the changes in the QEEG test possibly do correlate with prolonged consciousness/cognitive disorders and what the value of neurofeedback training is to these patients in respect to their improvement or, potentially, a complete recovery to the mental state pre-SARS-CoV-2 infection.

## 6. Discussion/Perspectives

The list of neurological symptoms potentially caused by the COVID-19 infection is long and includes headache, nausea, altered consciousness, strokes and the common loss or disturbance of the sense of smell and taste. There is growing evidence that these symptoms may result from the direct invasion of the CNS and from damage to certain structures or peripheral nerves by this virus [14,17,18,20,21,23,24,25,26,28]. A few case reports suggest that SARS-CoV-2 can cause seizures [25,26] while others describe non-specific encephalopathic patterns [13,15,16,17,18,19,22,23,27] or particular EEG changes, possibly characteristic for post-COVID-19 neuronal damage [18,22]. Therefore, it seems justified to introduce EEG testing in patients with COVID-19 both for diagnosing and monitoring the CNS affection by viral infection. The main challenge here is to differentiate neurotrophic related specific changes from those that possibly result from pulmonary or systemic deterioration observed in the course of COVID-19 or in existence before the infection. In particular, the risk of the overinterpretation of changes in brain function during the course of COVID-19 as virus-specific is quite high. It is clear that not only the neurotrophic properties of SARS-CoV-2 may result in encephalopathy. In fact, brain hypoxia, microemboli or just prolonged ICU treatment with long term sedation represent much more common causes for the impaired function of the brain that are caused by systemic complications of COVID-19 and not by the virus itself. The problem of non-specific symptoms, i.e., hypoxic encephalopathy, in the course of COVID-19 has been recognized and discussed in several of the cited reports [14,16,25]. Nevertheless, for the more systematic analysis of this issue, a study in a vast population of ICU patients implementing, for example, severity-matched groups analyzing COVID-19 subjects paired with non-infected controls would be desirable.

One important aspect in helping the discrimination between the specific and systemic impact of SARS-CoV-2 infection on the brain structure and function is the result of neuroimaging. In the analyzed literature, several studies referred to intact cerebral morphology indicated by unremarkable computer tomography (CT) findings as the sign of the purely functional COVID-19 related damage [17,19]. However, the limited specificity and spatial resolution of even a modern CT raises a question: if magnetic resonance imaging (MRI) is the method of choice for ruling out the structural changes in a SARS-CoV-2 affected brain. In particular, the possibility of demonstrating fresh or disseminated hypoxic damage is much higher using MRI compared with CT methods. Indeed, several reports include a description of MR images as the background for the interpretation of their (Q)EEG results. Interestingly, Pasini et al. as well as Galapoulou et al. declared that the majority of their COVID-19 patients displayed a normal MRI finding [13,18]. In contrast, Dixon et al. and De Stefano et al. reported cases in which EEG changes were associated with dramatic damage to the brain, presumably caused by the COVID-19 infection and clearly distinguishable in MR or even CT imaging [20,27]. Notably, in the studies analyzed, the lack of change in the cerebral structure was not necessarily associated with a good clinical condition [13,18]. Thus, it would be prudent to assess the impact of the SARS-CoV-2 infection on the EEG records in the context of structural brain damage (or its lack) visible in CT/MRI scans.

Another reasonable approach in rectifying the assessment of the impact of SARS-CoV-2 on EEG activity would be a correlation analysis of virus burden and the EEG spectrum, ideally as a multivariate analysis, in order to refine the impact of the virus itself from systemic disease related confounders. One important obstacle apart from the number of patients available for analysis is the heterogeneity of the EEG techniques used and certain subjectivity of their interpretation. Thus, the consequent implementation of QEEG techniques would enable a more precise, objective description of these changes and would create a solid base for attempts to implement QEEG neurofeedback therapy as a part of comprehensive treatment for patients who aspire to improve their neurological symptoms acquired in connection with the SARS-CoV-2 virus infection.

## Figures and Tables

**Figure 1 jcm-10-01300-f001:**
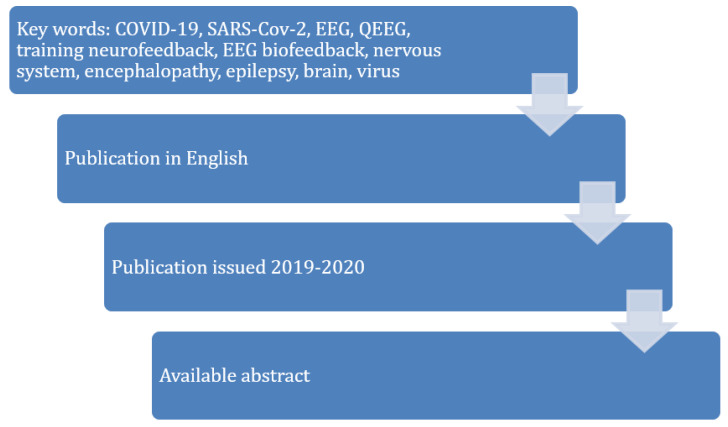
The stages of searching in the databases.

**Figure 2 jcm-10-01300-f002:**
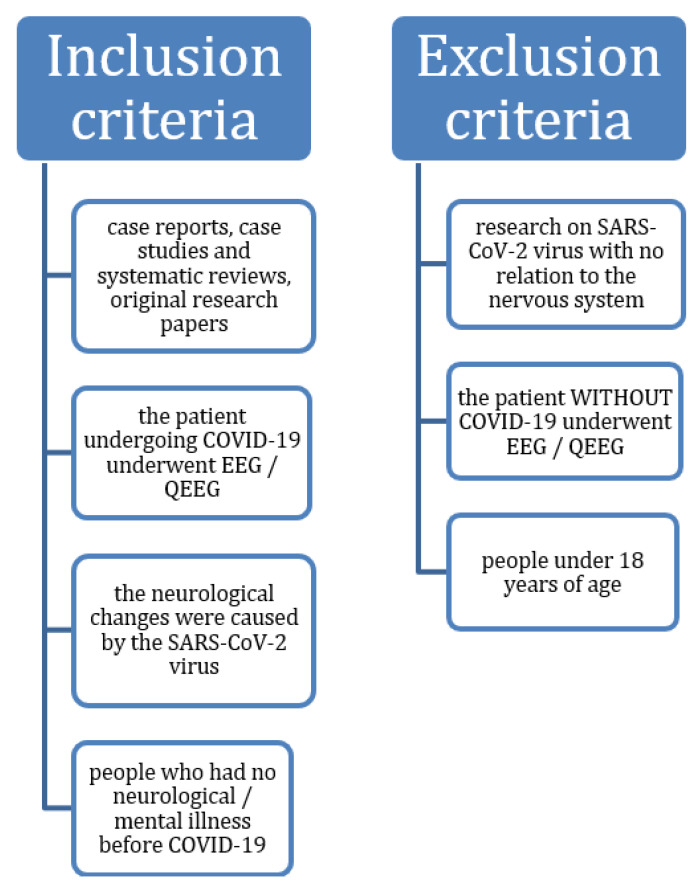
Inclusion and exclusion criteria applicable for the analysis.

**Table 1 jcm-10-01300-t001:** Characteristics of the studies included in the analysis.

Researchers	Aim	Materials and Methods	Results	Conclusions
Pasini E et al. (2020) (experimental research paper) [13]	Checking for the presence of an encephalopathy pattern in patients with COVID-19 with diagnosed neurological symptoms.	15 patients with suspected COVID-19 related encephalopathy.	EEG abnormalities similar to those found in patients with encephalopathy.	The SARS-CoV-2 virus can cause EEG changes; however, more research is needed.
Vellieux G et al. (2020) [14] (experimental research paper)	Checking whether the EEG record in patients infected with the SARS-CoV-2 virus manifests changes in the form of a new, unprecedented pattern.	A case report of two COVID-19 patients with neurological symptoms.	The EEG examination showed a new, unprecedented pattern.	EEG changes in patients with COVID-19 may be specific to their disease; however, more research is needed.
Pastor J et al. (2020) (experimental research paper) [15]	Using a QEEG to determine the characteristics of encephalopathy in patients with COVID-19.	A QEEG study was performed. The study group consisted of 20 hospitalized patients who had COVID-19. They were compared with two control groups.	Changes that were similar to the abnormalities found in encephalopathy were seen in patients who had a traumatic time with COVID-19.	The SARS-CoV-2 virus can cause neurological changes that are visible in the QEEG record.
Koutroumanidis M et al. (2020) (experimental research paper) [16]	Checking if COVID-19 related neurological changes are caused by viral neurotropism or due to hypoxia in the course of the disease.	An analysis of the encephalograms of 19 patients who underwent COVID-19.	Severe encephalopathy was found in 13 patients in the EEG.	A relatively high frequency of the rare alpha pattern may reflect direct SARS-CoV-2 neurotropism.
Flamand M et al. (2020) (experimental research paper) [17]	Check for EEG changes related to the SARS-CoV-2 virus.	A case report of an 80-year-old patient suffering from COVID-19 who underwent several EEG tests during her hospitalization.	The EEG record showed the development of three-phase waves.	EEG changes indicated a progressive neurological process that was possibly associated with SARS-CoV-2. This case indicated that more attention should be paid to the EEG patterns in patients during the COVID-19 pandemic.
Galanopoulou, A.S et al. (2020) (experimental research paper) [18]	Checking the neurological changes appearing in the EEG record in patients infected with the SARS-CoV-2 virus.	Twenty six adults were examined with an EEG (20 positive for the SARS-CoV-2 virus, six negative).	EEG changes/patterns similar to those in epilepsy appeared in 40.9% of COVID-19 positive patients. Changes with dominant frontal brain sharp waves were noticed in the record.	Future research must determine if COVID-19 infection increases the risk of the epilepsy-like abnormalities and investigate their pathogenesis.
Chen W et al. (2020) (experimental research paper) [19]	Assessment of EEG abnormalities in COVID-19 patients and assessment of epilepsy-like activity and seizures.	Five critically ill adults with COVID-19 underwent EEG monitoring.	All EEG recordings showed non-specific markers of encephalopathy as well as diffuse slowing and generalized rhythmic delta activity. Two patients also had epileptic discharges ranging from 2–3 Hz. The improvement in the EEG record and clinical symptoms was followed with anti-epileptic drugs.	The results showed the importance of EEG monitoring in COVID-19 patients and showed the positive effect of anti-epileptic drugs. However, more research is needed on a larger study group.
De Stefano P et al. (2020) (experimental research paper) [20]	Description of a patient suffering from COVID-19-associated acute respiratory syndrome (ARDS) to highlight the diagnostic role of the EEG in the ICU (intensive care unit).	A patient with COVID-19 underwent mechanical ventilation due to acute respiratory distress syndrome (ARDS) and had an altered mental state in the ICU. An EEG, magnetic resonance imaging and an analysis of the cerebrospinal fluid were performed.	A video EEG revealed a focal monomorphic theta slowing down in the bilateral fronto-central regions. In line with the localization of the EEG, MRI showed numerous microbleeds located in the bilateral junction of white matter and different areas of the corpus callosum and the inner capsule suggesting the presence of cerebral microhemorrhage. A cerebrospinal fluid analysis ruled out the presence of encephalitis.	An EEG allowed the detection of neurological dysfunctions in the ICU in a situation where it was difficult due to the severity of the respiratory ailments.
Cecchetti G et al. (2020) (experimental research paper) [21]	Presentation of neurological changes in COVID-19 patients and changes in the EEG recording.	Eighteen COVID-19 patients.	The presence of slow waves in the front in the EEG record was common. Lower values of oxygen saturation on admission were associated with more serious EEG abnormalities suggesting that higher level hypoxemia and possibly longer periods without treatment may contribute to brain dysfunction.	The EEG can be a useful tool for assessing early brain involvement in COVID-19 especially in severe cases.
Antony AR et al. (2020) (systematic review paper) [22]	A systematic literature review was carried out to synthesize data on EEG scores in COVID-19.	The available data from 617 patients were analyzed with the EEG results reported in 84 studies.	The occurrence of abnormal EEG patterns in COVID-19 patients.	EEG abnormalities were common in COVID-19 related encephalopathy and correlated with disease severity, pre-existing neurological conditions including epilepsy and prolonged EEG monitoring. Pre-frontal changes were common and were proposed as a biomarker for COVID-19 encephalopathy.
Vespignani H et al. (2020) (experimental research paper) [23]	Assessment of unexplained altered mental state, loss of consciousness or poor agitation and sensitivity in critically ill COVID-19 patients.	Twenty six critically ill hospitalized patients infected with SARS-CoV-2 underwent electroencephalography to evaluate an unexplained altered mental state, loss of consciousness or poor agitation and sensitivity.	Of the 26 patients tested, five patients had electroencephalograms that showed periodic discharges consisting of frontal brain monomorphic high amplitude delta waves in the absence of epileptic activity.	These findings could suggest central nervous system damage potentially related to COVID-19 in these patients.

## Data Availability

The original contributions presented in the study are included in the article; further inquiries can be directed to the corresponding author.

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
