# Peer review of "Changes in EEG Recordings in COVID-19 Patients as a Basis for More Accurate QEEG Diagnostics and EEG Neurofeedback Therapy: A Systematic Review"

_jcm, 2021, doi:10.3390/jcm10061300_

Round 1
Reviewer 1 Report
Introduction:
- You mention QEEG. Can you cite or refer a paper about the definition of QEEG?
- You mention "where the record from EEG sensors is quantified by specialized softwares". Can you cite or refer papers related to this method/softwares?
- Basically in the current paper draft, your Introduction does not cite or refer a paper at all. So it is not clear which are your opinions and which are definition or explanation from cited papers.
Table 1 and Table 2:
- Table 1 and Table 2 show your literature selection methods quite well. However, you may consider using graphics or flow charts or figures, instead of using tables. A picture is worth a thousand words. It would be lovely for the readers.
Table 3:
- In the first column "Researchers", can you check for the inconsistencies? (Sometimes you use author name and sometimes you use institution name)
- In the first column "Researchers", can you cite to your reference below?
- Can you separate in Table 3 which literatures are experimental research papers and which ones are review papers?
Discussion:
- Can you check the inconsistencies of which sentence cite which paper in the reference?
- You mention "In one of the reports, specialists from the
International Federation of Clinical Neurophysiology presented the EEG results of a group of 15 COVID-19 patients with suspected encephalopathy", but this sentence has no citation to your reference below. It is hard for the reader to know which citation related to which sentence. - You mention " [6] Another report presented by the
International Federation of Clinical Neurophysiology describes two cases of COVID-19 patients in
whom the EEG test showed a new, unprecedented pattern." and then in the end of next sentences you cite "[7]" It is not clear if this sentence cites paper [6] or [7]. - There are many more citations similar to above. Please check which sentence related to which citation to your reference.
Author Response
Dear Reviewer 1,
Thank you very much for your kind response with precious advices. We improved our manuscript with respect to your comments. We've changed table 1 and 2 into flow charts and re-organized table 3. We've also add description abour kind of paper (research paper, systematic review etc.) in table 3. In the discussion, we've changed the references and some partf of decsription. We hope, it's more clear now. All chenges are in red. I believe that we have answered all the comments adequately and have revised the manuscript properly, producing a more balanced and better account of our work. Thank You in advance for your kind consideration in this matter.
Yours sincerely,
Marta Kopańska

Reviewer 2 Report
As the authors note, there is a small but growing body of evidence that COVID-19 leads to neurological sequelae in some patients, and this is a potentially useful review of existing knowledge on the electrophysiological alterations due to the disease. While there is as yet still a paucity of published information, adding additional clinical and demographic information about the patients would facilitate in interpreting the overall results of this paper.
Major suggestions:
- First, what were the age ranges and median ages of the patients in the reported studies? Presumably, older patients would be more susceptible to complications.
- What other prior health problems did these patients possess? Importantly, were they hypoxic during the disease course?
- What were the imaging findings, if any, in these patients? Certainly, some neural alterations could be due to small strokes, inflammation, etc.
- Were the neurological complications from COVID any different from what is seen in other serious diseases, such as influenza?
- While COVID may directly invade the nervous system, direct evidence is not presented here. The findings that the authors present may be due to strokes, inflammation, hypoxia, etc., as a result of the infection.
Minor suggestions
1. Some of the text in the tables was cut off in my document.
Author Response
Dear Reviewer 2,
Thank you very much for your kind response with precious advices. We improved our manuscript with respect to your comments. We've changed table 1 and 2 into flow charts and re-organized table 3. We've also add description abour kind of paper (research paper, systematic review etc.) in table 3. In the discussion, according you your advices, we've add age ranges and median ages of patients. We've tried to answer the remaining questions. More than once, there was not much information about the patients, so we included as much as we could. We've also improved our English in the document. We hope, it's more clear now. All changes are in red. I believe that we have answered all the comments adequately and have revised the manuscript properly, producing a more balanced and better account of our work. Thank You in advance for your kind consideration in this matter.
Yours sincerely,
Marta Kopańska

Round 2
Reviewer 1 Report
General comment:
- It is an improvement from the previous version.
- Some misspellings can be fixed, by turning on the spelling checker if you are using Microsoft Word.
- I suggest you use citation like "... of the cerebral cortex [1]. " or "... based on the test-retest method [2,3]."
The punctuation is after the citation, not before the citation. - There are examples how to cite, as well as how to make tables and figures, in this journal, for example Alateeq, et al, 2021 (https://doi.org/10.3390/jcm10040637) or Neal, et al, 2021 (https://doi.org/10.3390/jcm10040604).
You can also check the paper on your citation Pastor, et al, 2020 [20] - Make consistent use of "qEEG" or "QEEG"
Figure 2:
Make sure the words in the figure are readable, by choosing better font.
Table 1:
- Who are Aristea S et., al? You mention on the table, but not in the reference.
Discussion:
- Last sentence of paragraph 1 "... SARS-CoV-2 may cause permanent damage to certain areas of the CNS or peripheral nerves [13, 14, 15, 16]".
Can you explicitly write CNS here Central Nervous System (CNS)? (definition of abbreviation, for better readibility)
Reference:
reference [5] and [12] are the same: Kreper, et al, 2020, "A multicenter effectiveness trial of QEEG-informed neurofeedback in ADHD: Replication and treatment prediction".
Can you check again the citation redundancies in the discussion?
"... the most appropriate individualized therapeutic protocol according to the patient's condition. [4, 5]"
"... attention deficit hyperactivity disorder. [12] "
Author Response
Dear Reviewer 1,
the Authors greatly appreciate the valuable advices. The Authors have corrected minor linguistic errors within the text as well as pointed citations. The Authors have standardized the font to make the graphics legible. Thank to these corrections, our manuscript became improved. All changes are in red.
Thank you very much for so useful advices.
Best regards, Marta Kopańska
Reviewer 2 Report
The paper is much improved since the initial draft and is much better referenced. I do still have some questions about the reviews they've performed, although the data may not be available.
- In the papers you cite, were there confounding factors such as hypoxia or stroke that may have affected the neurological status in these patients? This is a crucial piece of information against which to evaluate the effects of the virus itself against common sequelae of severe respiratory infections.
- Are there any imaging findings that can add more context to the data? While this is an electrophysiological paper, I have heard anecdotally that some patients with COVID do not have obvious structural findings upon imaging. This would also assist with the interpretation of the EEG results.
Minor changes- English editing would help with readability. Also, change "Discussion" to "Results", and possibly change "Conclusion" to Discussion.
Author Response
Dear Reviewer 2,
the Authors greatly appreciate the valuable advices. The Authors have corrected minor linguistic errors within the text as well as pointed citations. We also tried to answer all questions and doubts. Now, in our opinion - thanks to your advices, our manuscript is full of important information in this field. Thank to these corrections, our manuscript became improved. All changes are in red.
Thank you very much for so useful advices.
Best regards, Marta Kopańska